



# Accelerating Bayesian microseismic event location with deep learning

Alessio Spurio Mancini[1,2], Davide Piras[1], Ana Margarida Godinho Ferreira[3], Michael Paul Hobson[2], and Benjamin Joachimi[1]

[1]Department of Physics and Astronomy, University College London, Gower Street, London, WC1E 6BT, UK
[2]Astrophysics Group, Cavendish Laboratory, J. J. Thomson Avenue, Cambridge, CB3 0HE, UK
[3]Department of Earth Sciences, Faculty of Mathematical  Physical Sciences, University College London, WC1E 6BT, United Kingdom

**Correspondence:** Alessio Spurio Mancini (a.spuriomancini@ucl.ac.uk)

**Abstract.** We present a series of new open source deep learning algorithms to accelerate Bayesian full waveform point source inversion of microseismic events. Inferring the joint posterior probability distribution of moment tensor components and source location is key for rigorous uncertainty quantification. However, the inference process requires forward modelling of microseismic traces for each set of parameters explored by the sampling algorithm, which makes the inference very computationally

intensive. In this paper we focus on accelerating this process by training deep learning models to learn the mapping between source location and seismic traces, for a given 3D heterogeneous velocity model, and a fixed isotropic moment tensor for the sources. These trained emulators replace the expensive solution of the elastic wave equation in the inference process.

We compare our results with a previous study that used emulators based on Gaussian Processes to invert microseismic events. For fairness of comparison, we train our emulators on the same microseismic traces and using the same geophysical setting.

We show that all of our models provide more accurate predictions and $\sim 100$ times faster predictions than the method based on Gaussian Processes, and a $\mathcal{O}(10^5)$ speed-up factor over a pseudo-spectral method for waveform generation. For example, a 2-s long synthetic trace can be generated in $\sim 10$ ms on a common laptop processor, instead of $\sim 1$ hr using a pseudo-spectral method on a high-profile Graphics Processing Units card. We also show that our inference results are in excellent agreement with those obtained from traditional location methods based on travel time estimates. The speed, accuracy and scalability

of our open source deep learning models pave the way for extensions of these emulators to generic source mechanisms and application to joint Bayesian inversion of moment tensor components and source location using full waveforms.

## 1  Introduction

The monitoring of microseismic events is crucial to understand induced seismicity and to help quantify seismic hazard caused by human activity (Mukuhira et al., 2016). Accurate event locations are key to map fracture zones and failure planes, ultimately

enhancing our understanding of rupture dynamics (e.g. Baig and Urbancic, 2010).

Seismic inversion for earthquake location has traditionally been based on the minimisation of a misfit function between theoretical and observed travel times (see e.g. Wuestefeld et al., 2018, for a review of different methods applied to microseismic





events). These optimisation-based methods are essentially refinements of the original iterative linearised algorithm proposed by Geiger (1912), focusing on improving the misfit function or the optimisation technique (see, e.g., Li et al., 2020, for a

comprehensive review). Since the 1990s, non-linear earthquake location techniques have been developed using, e.g. the genetic algorithm (Kennett and Sambridge, 1992; Šílený, 1998), Monte Carlo algorithms (Sambridge and Mosegaard, 2002; Lomax et al., 2009) and grid searches (e.g., Nelson and Vidale, 1990; Lomax et al., 2009; Vasco et al., 2019). The majority of these methods uses arrival times and require phase picking. Recently, waveform-based methods have emerged, such as waveform stacking (e.g. Pesicek et al., 2014) or time reverse imaging (e.g. Gajewski and Tessmer, 2005), which do not only consider

arrival times, but also use other information from the waveforms. Full waveform inversion methods, which are based on the comparison between simulated full synthetic waveforms and observations, are also being increasingly used to enhance the determination of event locations (Kaderli* et al., 2015; Behura, 2015; Cesca and Grigoli, 2015; Wang, 2016; Shekar and Sethi, 2019).

Bayesian inference has been successfully used to locate earthquakes and to estimate moment tensors (e.g., Tarantola, 2005;

Wéber, 2006; Lomax et al., 2009; Mustać and Tkalčić, 2016). Within the Bayesian framework, the ultimate goal is to provide estimates of the posterior distribution of the model parameters (see, e.g., Xuan and Sava, 2010, for an application to microseismic activity). Bayesian source inversions use techniques such as Markov Chain Monte Carlo (MCMC) to sample the parameters' posterior distribution (see e.g. Craiu and Rosenthal, 2014, for a review of different sampling algorithms). They allow the rigorous inclusion and propagation of different uncertainties, such as those arising from the assumed velocity model

for the seismic domain that is being studied (see e.g. Pugh et al., 2016b). So far Bayesian source inversions for locations and moment tensors have been typically performed using travel time measurements (Lomax et al., 2009) or amplitude and polarity data (Pugh et al., 2016a, b; Pugh and White, 2018). Here we carry out Bayesian source location inversions of microseismic events using the full waveform information.

Ideally, the inversion could be carried out jointly for the moment tensor components and the location of the microseismic

event (Rodriguez et al., 2012; O'Toole, 2013; Käufl et al., 2013; Stähler and Sigloch, 2014; Li et al., 2016; Pugh et al., 2016b; Pugh and White, 2018; Willacy et al., 2019). However, when using full waveforms this is extremely computationally intensive. While performing MCMC sampling, the forward model needs to be simulated at each point in parameter space where the likelihood function is evaluated. The number of such evaluations scales exponentially with the number of parameters (an example of the curse of dimensionality, see e.g. MacKay 2003). Since the solution of the elastic wave equation for forward modelling

microseismic traces in complex media is computationally very expensive, this means that, even for small parameter spaces, sampling the posterior distribution becomes extremely challenging or even unattainable. For example, given the geophysical model with microseismic activity considered in Das et al. (2017), i.e., a 3D heterogeneous velocity model on a 1 km × 1 km × 3 km grid, the generation of a single seismic trace with a pseudo-spectral method (Treeby et al., 2014) for a given source requires $\mathcal{O}(1)$ hour of Graphics Processing Unit (GPU) time with a Nvidia P100 GPU. Using typical MCMC methods, this

operation may need to be repeated for tens or hundreds of thousands of points in parameter space, to ensure convergence of the sampling algorithm.





To overcome this issue, Das et al. (2018, referred as D18 hereafter) developed a machine learning framework (also referred to as metamodel, surrogate model or emulator) for fast generation of synthetic seismic traces, given their locations in a marine domain and a specified 3D heterogeneous velocity model, and a fixed isotropic moment tensor for all the sources. Gaussian
Processes (GPs, Rasmussen and Williams, 2005) were trained as surrogate models that could be employed for Bayesian inference of microseismic event location (with fixed isotropic moment tensor) to replace the expensive solution of the elastic wave equation for each set of source coordinates explored in parameter space. Other recent studies have also used deep learning approaches for fast approximate computations of synthetic seismograms (e.g. Moseley et al., 2018; Moseley et al., 2020a, b) and for earthquake detection and location (e.g. Perol et al., 2018).

In this paper, we build on the method developed in D18 by training multiple generative models, based on deep learning algorithms, to learn to predict the seismic traces corresponding to a given source location, for fixed moment tensor components. Similar to D18, we consider an isotropic moment tensor for our sources; a follow-up paper (Piras et al., 2021) extends the methodologies proposed here to different source mechanisms (compensated linear vector dipole and double-couple, see Sec. 4 for a discussion). Once trained, our generative models can replace the forward modelling of the seismograms at each likelihood
evaluation in the posterior inference analysis. We show that the newly proposed generative models are more accurate than the results of D18. In addition, the emulators we develop are faster by a factor of $\mathcal{O}(10^2)$, less computationally demanding and easier to store than the D18 surrogate model. We also demonstrate how our new emulators make it possible in practice to perform Bayesian inference of a microseismic source location. We validate our results by carrying out a comparison of our results with a common nonlinear location method based on travel time estimates (Lomax et al., 2000).

In Sec. 2 we present our generative models and the general emulation framework. We first describe the preprocessing operations operated on the seismograms to facilitate the training of our new generative models, which are subsequently outlined in detail together with general notes on their training, validation and testing. In Sec. 3 we apply the emulation framework to the same test case studied by D18, and we compare the results achieved by the different methodologies. We also use our best performing model to show that we can accelerate accurate Bayesian inference of a simulated microseismic event, and compare
the estimated source location with that retrieved by a standard nonlinear location method. Finally, we conclude in Sec. 4 with a discussion of our main findings and their future applications.

## 2  Generative models

In this Section we describe the deep generative models that we train as emulators of the seismic traces given their source location. Our final goal is to develop fast algorithms that can learn the mapping between source location and seismic traces
recorded by receivers in a geophysical domain.

We start in Sec. 2.1 describing the preprocessing operated on the seismic traces for feature selection and dimensionality reduction. We then describe the algorithms for emulation of the preprocessed seismic traces in Sec. 2.2. These methods are machine learning algorithms that can, in principle, be applied to seismograms recorded in any geophysical scenario; in fact, these algorithms have been applied to areas beyond geophysics (see e.g. Auld et al., 2007, 2008, for applications to cosmology).



While we initially present our emulators without referring to any particular geophysical scenario, for concreteness we also present a choice of the methods' hyperparameters (e.g. the number of layers and nodes of the neural networks employed) based on their application to the test case later described in Sec. 3. We report these specific hyperparameter choices to provide an example of a practical successful implementation of the machine learning algorithms, but we stress again that the generative models presented in Sec. 2.2 are applicable to any geophysical scenario, provided enough representative training samples and a

velocity model are available. They may, however, require different hyperparameter choices, depending on the specific domain considered. As discussed in more detail in Sec. 3.2, this hyperparameter tuning is not computationally expensive because our models are very easy to train.

  Training, validation and testing procedures for our generative models are described in Sec. 2.3, with emphasis on the metrics used to compare the accuracies of the different algorithms. We note that the number of training, validation and testing samples

required for each method may vary according to the specific geophysical domain considered; for example, larger geophysical domains may require a larger number of training samples, reflecting the increased variability in the seismic traces. Once again, in Sec. 2.3 we discuss a training procedure that has produced successful results on the test case considered in Sec. 3. Applications to different geophysical scenarios may need slightly different tuning of the hyperparameters involved in the training procedure, but the general technique shown in 2.3 can be easily adapted to incorporate these changes.

## 2.1 Preprocessing

In order to train fast emulators to replace the simulation of microseismic traces for a given source location we need to generate representative examples of the seismograms to be learnt, given a fixed velocity model for the geophysical scenario considered. The complexity of the forward modelling of seismic traces by means of e.g. pseudo-spectral methods (see e.g. Faccioli et al., 1997) implies that only a relatively small number of training samples can realistically be generated. In turn this means that the

emulation of seismic traces by means of even just a simple neural network (which will be described in Sec. 2.2.1) will only lead to overfitting the training set, as we verified directly. This issue can be relieved by applying some form of preprocessing to the data, in order to reduce the number of relevant features that have to be learnt by the emulators. In addition, some compression method can be employed to reduce even further the dimensionality of the mapping, on condition that the performed compression is efficient in preserving the information carried by the original signal.

To preprocess our seismograms we first identify the maximum positive amplitude $A_i$ and the corresponding time index $t_i$ in each seismogram, labelled by index $i = 1, ..., N_{\text{train}}$, in our training set of $N_{\text{train}} = 2000$ samples (the same used by D18). We then isolate one random seismic event in our training set and store the value of its maximum positive amplitude $A^*$ and its corresponding time index $t^*$. We normalise all of our training seismograms to the amplitude of this reference peak, and shift them so that their peak location corresponds to the reference peak location (see Fig. 1), replacing the missing time components with zeros. Operating this preprocessing leaves us with two additional parameters for each seismic trace: a normalising factor

$\bar{A}_i \equiv A_i/A^*$ and a time shift $\bar{t}_i \equiv t_i - t^*$. This preprocessing is encouraged by the structure of the signal, which is localised in the form of spikes preceded by absence of signal, corresponding to the sudden arrival of the P-wave at the sensor location. The amplitudes $A_i$ and time indices $t_i$ depend mainly on the distance of the seismic source from the sensor. By rescaling all





training seismograms to the reference amplitude $A^*$ and time index $t^*$ we allow the deep learning algorithm to 'concentrate'
on learning the rest of the signal, which instead depends on the properties of the heterogeneous medium encountered by the
wave while propagating to the sensor. We verified that all numerical conclusions of our analysis do not depend significantly
on the specific choice of the reference seismogram; therefore, in our analysis we choose the reference seismogram completely
at random. It could be argued that the reference seismogram should be chosen in order to minimise the truncated signal, i.e.,
that for every receiver we should choose the closest event as the reference seismogram. However, we verified that, in order to
constrain the source location adequately well, the main peak of a seismogram is sufficient, and therefore this choice is irrelevant
in our approach.

A consequence of this type of preprocessing is that, in order to recover the original seismograms, one also needs to learn
the coefficients $\bar{A}_i$ and $\bar{t}_i$. To this purpose we train two additional independent machine learning algorithms that map the input
source coordinates $(x_i, y_i, z_i, d_i)$ to the output amplitude and time shift $(\bar{A}_i, \bar{t}_i)$. We include the distance $d_i$ from the receiver
in the set of coordinates, as we observed that it improves the emulation performance). We model each of these outputs as a
Gaussian process (GP, Rasmussen and Williams, 2005). When performing GP regression given a generic function $f(\boldsymbol{\theta})$ of
parameters $\boldsymbol{\theta}$, we assume

$$f(\boldsymbol{\theta}) \sim \mathcal{N}\left(0, K(\boldsymbol{\theta}, \boldsymbol{\theta}'; \boldsymbol{\psi})\right) , \tag{1}$$

where the kernel $K(\boldsymbol{\theta}, \boldsymbol{\theta}'; \boldsymbol{\psi})$ represents the covariance between two points in parameter space and may depend on additional
hyperparameters, collectively denoted as $\boldsymbol{\psi}$. In our case $\bar{A}$ and $\bar{t}$ are modelled as functions of the coordinates $(x, y, z, d)$ using
a GP each. For the geophysical domain studied in Sec. 3 we find that a Matérn kernel $K$ in its Automatic Relevance Discovery
(ARD) version (Neal, 1996; Rasmussen and Williams, 2005), defined as

$$K_{\mathrm{ARD-Mat\acute{e}rn}-3/2}(\boldsymbol{\theta}, \boldsymbol{\theta}'; \boldsymbol{\psi}) = \sigma_f^2 \left(1 + \sqrt{3}\tilde{r}\right) \exp\left(-\sqrt{3}\tilde{r}\right) , \tag{2}$$

where

$$\tilde{r} = \sqrt{\sum_{m=1}^{n} \frac{(\theta_m - \theta_m')^2}{\sigma_m^2}} , \tag{3}$$

produces the most accurate results in training both GPs. The hyperparameters of this kernel are a signal standard deviation
$\sigma_f$ and a characteristic length scale $\sigma_m$ for each input feature $m = 1, \ldots, n$ (the source location, in our case). We optimise
these parameters while training our GPs, which we implement using the software GPY. Fig. 2 presents the general emulation
framework, showing in particular how the $\bar{t}$ and $\bar{A}$ produced by the trained GPs described above combine with the emulated
preprocessed seismogram to produce the final emulation output.

## 2.2  Machine learning algorithms

Here we present in detail the algorithms we developed for emulation of the seismic traces. Given a set of coordinates $x, y, z, d$,
each method outputs a seismogram preprocessed following the procedure described in Sec. 2.1. This means that for each



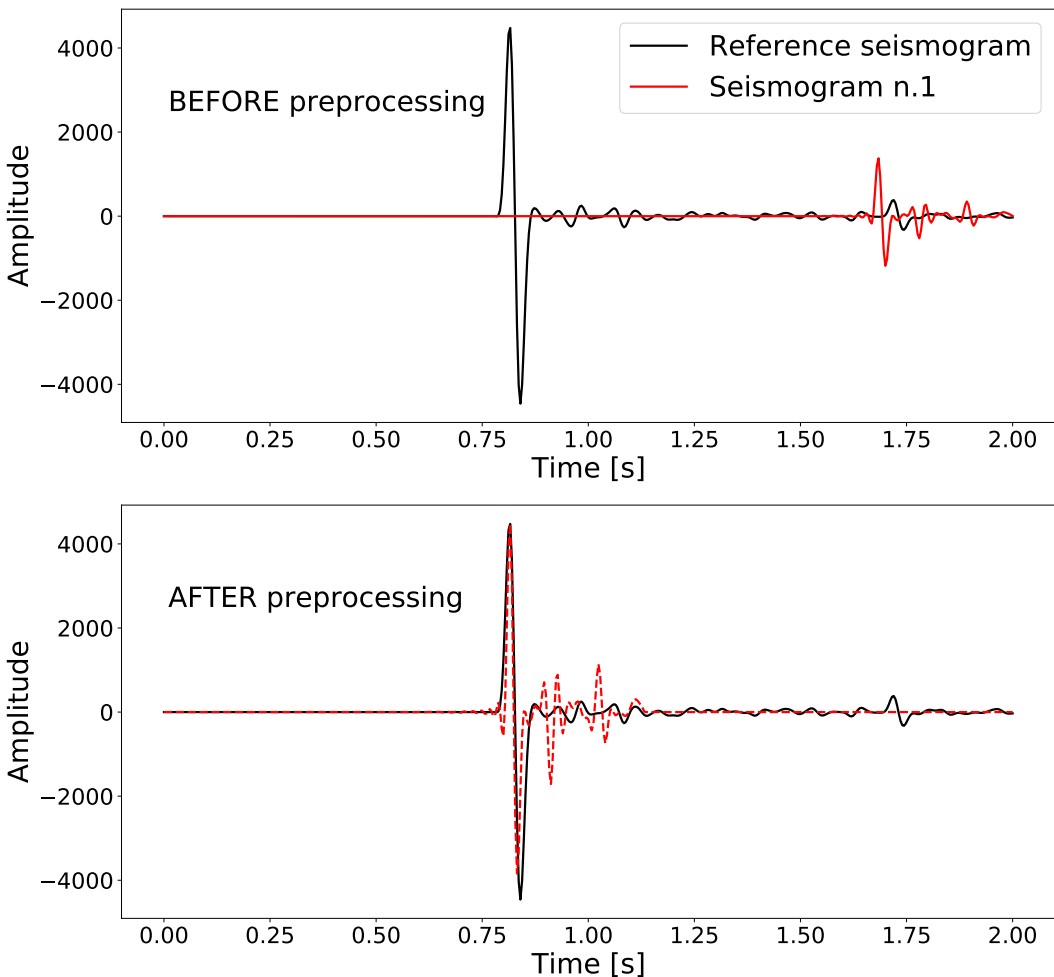

**Figure 1.** Example of preprocessing applied to the seismograms. We consider one random reference seismogram, shown in *black* in both upper and lower panels. Given another generic seismogram (*red* line in the upper panel), we rescale it to have its positive maximum peak amplitude and time location matching those of the reference seismogram. The result is a seismogram, like the one shown in *red* in the bottom panel, whose main difference with the reference seismogram is given by the additional fluctuations surrounding the main peak. The generative methods we develop learn to predict these fluctuations given the source location as well as the main peak, whereas two GPs learn the amplitude and time shift coefficients to rescale the predicted seismograms back to their natural amplitude and peak location. Throughout the paper, the seismograms' amplitude is measured in arbitrary units of pressure.





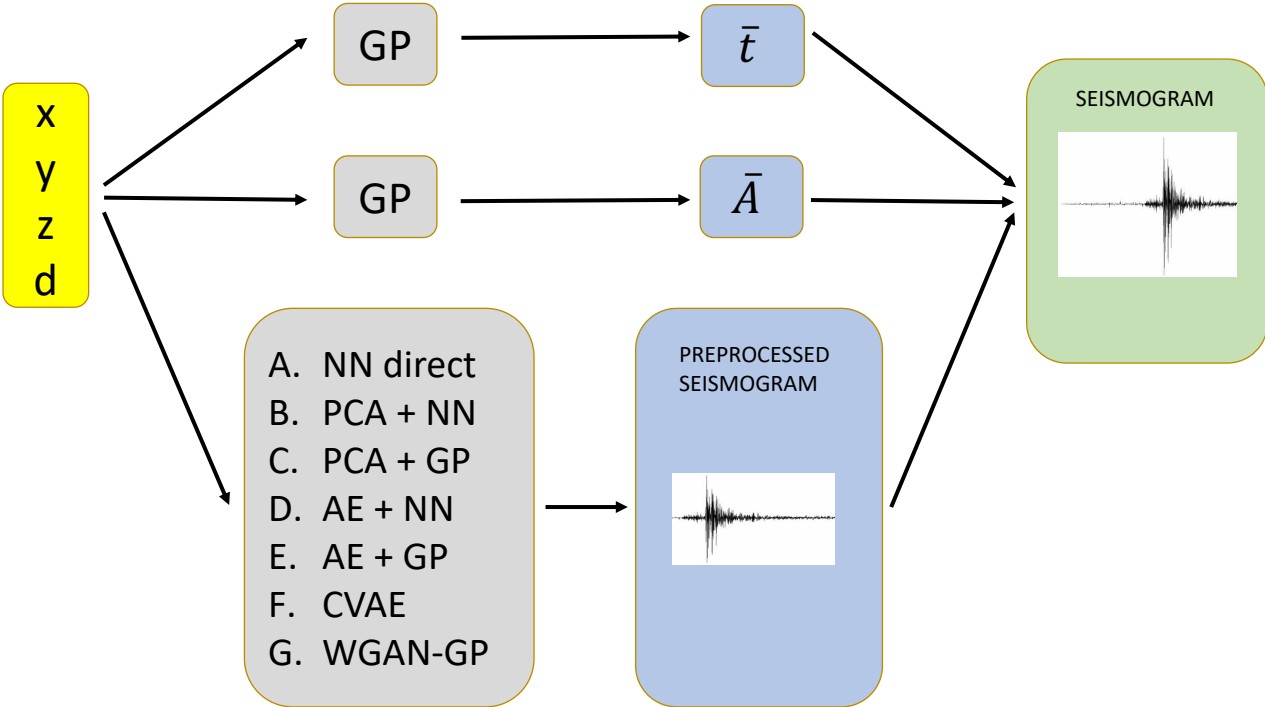

**Figure 2.** Schematic of the generic framework for seismograms emulation developed in this work. Two Gaussian Processes (GP) are trained to learn the preprocessing parameters $\bar{t}$ and $\bar{A}$, described in Sec. 2.1. One out of seven algorithms (described schematically in Fig.3 and in detail in Sec. 2.2) is chosen to generate a preprocessed seismogram. Finally, the combination of this learnt preprocessed seismogram and the learnt $\bar{t}$ and $\bar{A}$ gives the output seismogram corresponding to the coordinates $(x, y, z, d)$ (we augment the spatial coordinates $(x, y, z)$ with the distance $d$ from the receiver, since we noticed that it improves the accuracy of the trained models).

method, we also need to train two GPs to learn the mapping between source location and the coefficients $(\bar{A}, \bar{t})$, as described

in Sec. 2.1. In Fig. 3 we provide a schematic summarising the mapping between source coordinates and seismograms for each emulation framework described below. The specific architectures and hyperparameters represented in Fig. 3 are those optimised for the application described in Sec. 3.

### 2.2.1 Direct neural network mapping between source location and seismograms ('NN direct')

The first method we propose is a simple direct mapping between source location and preprocessed seismograms, without any

intermediate data compression. The mapping is learnt by a fully connected neural network, which consists of a stack of layers, each made of a certain number of neurons. Each layer maps the input of the previous layer $\boldsymbol{\theta}_{\text{in}}$ to an output $\boldsymbol{\theta}_{\text{out}}$ via

$$\boldsymbol{\theta}_{\text{out}} = \mathcal{A}\left(\boldsymbol{w}\,\boldsymbol{\theta}_{\text{in}} + b\right) \ , \tag{4}$$





**Figure 3.** Schematic of the seven proposed algorithms to learn the mapping between coordinates $(x, y, z, d)$ and preprocessed seismograms. A neural network (NN) is used in method (A), connecting directly source location to preprocessed seismograms. In methods (B) and (C) the preprocessed seismograms of the training set are compressed in Principal Component Analysis (PCA) coefficients, which are then learnt by a NN and Gaussian Processes (GPs), respectively. In method (D) and (E) the seismograms are compressed in central features of an autoencoder (AE), which are then learnt by a NN and a GP, respectively. Finally, a Conditional Variational Autoencoder (CVAE) and Wasserstein Generative Adversarial Networks - Gradient Penalty (WGAN-GP) are used in method (F) and (G), respectively, to learn the mapping between source location and preprocessed seismograms. In the schematic, the number of GPs and the architectures of the NNs (including the number of nodes for each layer, represented by blocks of different size, and the linear/non linear activation functions represented by arrows) are the ones used for the application to the geophysical domain described in Sec. 3.





where $\boldsymbol{w}$ and $b$ are called the network weights and bias, respectively, and $\mathcal{A}$ is the activation function, which is introduced in order to be able to model non-linear mappings. The output of each layer becomes the input to the following layer, and the number of neurons in each layer determines the shape of $w$ and $b$. Training the neural network consists of optimising the weights and biases to minimise a specific loss function which quantifies the deviation of the predicted output from the target training sample. The optimisation is performed by back-propagating the gradient of the loss function with respect to the networks parameters (Rumelhart et al., 1988).

For the specific application considered in Sec. 3, after experimenting with different architectures and activation functions, we find our best results are achieved with a neural network made of three hidden layers, with 64, 128 and 256 hidden units each, and a Leaky ReLU (Maas, 2013) activation function for each hidden layer, except the last one where we maintain a linear activation. The Leaky ReLU activation function for an input $x$ is defined as:

$$f(x) = \begin{cases} x & \text{if } x > 0 \\ \alpha x & \text{otherwise} \end{cases} \tag{5}$$

where we set the hyperparameter $\alpha = 0.2$, and we use a learning rate of $10^{-3}$. The Leaky ReLU activation function is a variant of the Rectified Linear Unit activation function (ReLU), which improves on a limitation of the ReLU activation function, sometimes referred to as "dying ReLU", whereby large weight updates mean that the summed input to the activation function is always negative, regardless of the input to the network (Xu et al., 2015). This means that a node with this problem will forever output an activation value of 0. We verified experimentally that Leaky ReLU performs indeed better than ReLU and other common activation functions. We choose the mean squared error (MSE) as our loss function:

$$\text{MSE} = \frac{1}{N_\text{t}} \sum_{m=1}^{N_\text{t}} \left( \tilde{S}_m - S_m \right)^2 , \tag{6}$$

between predicted and original seismograms $\tilde{\boldsymbol{S}}$ and $\boldsymbol{S}$. Image (A) in Fig. 3 summarises the emulation framework with direct NN between source coordinates and preprocessed seismograms.

### 2.2.2 Principal Component Analysis compression + neural network ('PCA+NN')

The second method proposed makes use of a signal compression stage prior to the emulation step. We first perform Principal Component Analysis (PCA) of the preprocessed seismograms in the training set. PCA is a technique for dimensionality reduction performed by eigenvalue decomposition of the data covariance matrix. This identifies the principal vectors, maximising the variance of the data when projected onto those vectors. The projections of each data point onto the principal axes are the 'principal components' of the signal. By retaining only a limited number of these components, discarding the ones that carry less variance, one achieves dimensionality reduction. For example, in our application to the test case described in Sec. 3, we retain only the first $N_\text{PCA} = 20$ principal components, as we find that in this case the 2D correlation coefficient between original and reconstructed seismograms is $R_{2D} \sim 0.95$. We can then model the seismograms as linear combinations of the PCA basis





functions $\boldsymbol{f}_i$,

$$\boldsymbol{S}(x,y,z,d) = \sum_{i=1}^{N_{\mathrm{PCA}}} c_i(x,y,z,d)\boldsymbol{f}_i , \qquad (7)$$

where the coefficients $c_i(x,y,z,d)$ are unknown non-linear functions of the source coordinates. We train a NN to learn this

mapping. In other words, contrary to the direct mapping between coordinates and seismogram components, we train a NN to learn to predict the PCA basis coefficients $c_i$ given a set of coordinates. Image (B) in Fig. 3 summarises the emulation framework in this case. We find that a neural network architecture similar to the one employed in the direct mapping approach, with three layers and LeakyReLU activation function, performs well also for this task. The number of nodes in each hidden layer is reduced to 50, and we still minimise the MSE between predicted and original PCA coefficients.

### 200    2.2.3    Principal Component Analysis compression + Gaussian process regression ('PCA+GP')

Once PCA has been performed on the training set, as an alternative to a neural network one can train multiple GPs to learn the mapping between the source coordinates and the PCA coefficients. We train one GP for each PCA component. Image (C) in Fig. 3 summarises the emulation framework for this approach.

### 2.2.4    Autoencoder compression + neural network ('AE+NN')

An autoencoder (AE) is a neural network with equal number of neurons in the input and output layers, trained to reproduce the input in the output (Hinton and Salakhutdinov, 2006). An autoencoder is typically made of an encoder followed by a decoder. The encoder network maps the input signal into a central layer (latent space), usually with fewer neurons with respect to the input to achieve dimensionality reduction. The decoder network receives as an input the output layer of the encoder and learns to map these compressed features back to the original input signals of the encoder. Together, encoder and decoder form a

'funnel-like' structure for the AE network, as shown in Fig. 4. In seismology, autoencoders have been studied by Valentine and Trampert (2012), who used them to compress seismic traces, and they are generally used as a non-linear alternative to PCA.

Once the AE has been trained, the new input signals can be compressed into the central features of the AE. Our aim is to learn the mapping between the source coordinates and these features. For example, this can be achieved with an additional neural network. Once this NN is trained, it can be used to generate new encoded features of the AE from new coordinates,

decoding new features into preprocessed seismograms. This procedure is summarised in Image (D) in Fig. 3.

In our test case of Sec. 3, we find that a fully-connected architecture with 501, 256, 128, 64, 5 nodes for each layer in the encoder (from the input to the latent space, and symmetric decoder) with Leaky ReLU activation function produces the best results in compressing the seismograms. Hence, we encode our seismograms in $z_{\mathrm{dim}} = 5$ central features; using a higher number of central features does not lead to significant improvements in the reconstruction performance, as discussed in Sec. 3.2. We

experimented also with a convolutional architecture, but noticed that it did not yield better accuracy, while also slowing down the training considerably.



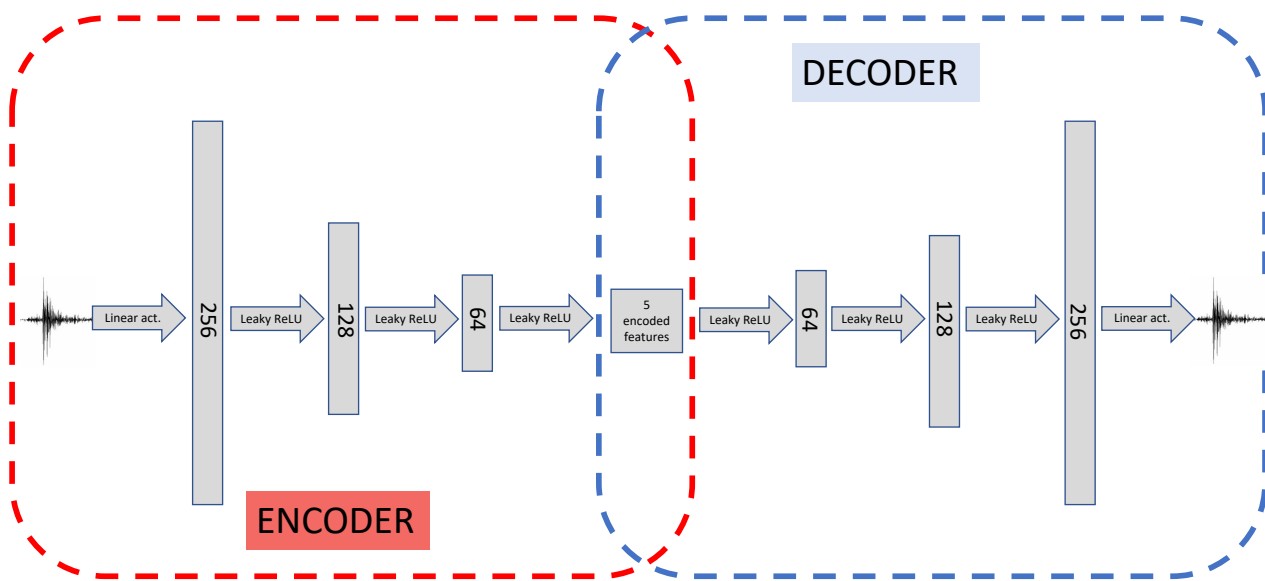

**Figure 4.** Typical architecture of an autoencoder. A bottleneck architecture allows for the compression of the input signal into a central layer through the 'encoder' part of the network (in *red*). The central layer is characterised by fewer nodes than the input one, thus leading to dimensionality reduction on condition that the 'decoder' part (in *blue*) can efficiently reconstruct the input signal (to a good degree of accuracy) starting from the central encoded features. In this schematic we highlight that training of the autoencoder is performed by feeding a seismogram to the encoder, and then comparing the output of the decoder with the same input seismogram. Once the autoencoder has been trained, the encoder can be removed and the decoder can be used as a generative model for the seismograms, inputting some encoded features.

### 2.2.5 Autoencoder compression + Gaussian process regression ('AE+GP')

Similarly to what we did with the PCA+GP method described in Sec. 2.2.3, one can train GPs to predict the encoded features given source coordinates. The predicted encoded features are then decoded by the trained decoder to generate new preprocessed 225 seismograms. The scheme is summarised in Image (E) in Fig. 3.

### 2.2.6 Conditional Variational Autoencoder ('CVAE')

In general, the encoded features in the latent space of an autoencoder have no specific structure, as the only requirement is for the reconstructed data points to be similar to the input points. However, it is possible to enforce a desired distribution over the latent space, which is driven by our preliminary knowledge of the problem and is therefore called a *prior* distribution. This is 230 one of the advantages of Variational Autoencoders (VAEs, Kingma and Welling, 2013). In this case, the model becomes fully



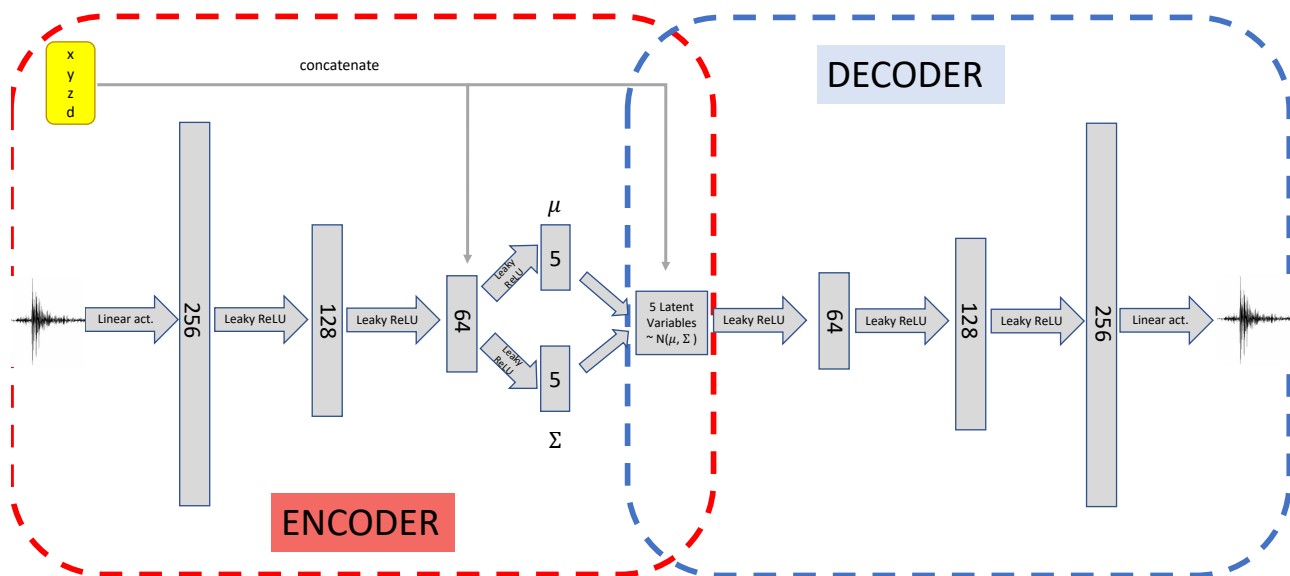

**Figure 5.** Schematic of the architecture of the Conditional Variational Autoencoder used in this work. A 'funnel-like' structure analogous to the simple autoencoder described in Fig. 4 is used, with the central features being sampled from multivariate Gaussian distributions $\mathcal{N}(\mu, \Sigma)$ with mean $\mu$ and covariance $\Sigma$, as described in Sec.2.2.6, after concatenating the coordinates $(x, y, z, d)$ to the last layer of the encoder (circled in *red*). The concatenation is repeated in the latent space represented by the multivariate Gaussian distributed encoded features. In a similar fashion to the simple autoencoder case, the decoder part (in *blue*) of the Conditional Variational Autoencoder can be used as a generative model, after training the full network to reproduce the input seismograms in output.

probabilistic, and the loss function to maximise consists of the ELBO (Evidence Lower BOund), which is defined as:

$$\text{ELBO} = \mathbb{E}_{\boldsymbol{z}}\left[\log p_\phi(\boldsymbol{x}|\boldsymbol{z})\right] - \text{D}_{\text{KL}}\left(q_\theta(\boldsymbol{z}|\boldsymbol{x})||p(\boldsymbol{z})\right) \ , \tag{8}$$

where $\boldsymbol{x}$ indicates the seismograms, $\boldsymbol{z}$ the encoded features, and $\mathbb{E}_{\boldsymbol{z}}$ the expectation value over $\boldsymbol{z} \sim q_\theta(\boldsymbol{z}|\boldsymbol{x})$. Additionally, $p(\boldsymbol{z})$ refers to the prior distribution we wish to impose in latent space, $q_\theta(\boldsymbol{z}|\boldsymbol{x})$ to the encoder distribution, and $p_\phi(\boldsymbol{x}|\boldsymbol{z})$ to

the decoder distribution; $\theta$ and $\phi$ indicate the learnable parameters of the encoder and of the decoder, respectively. Finally, $\text{D}_{\text{KL}}$ refers to the Kullback-Leibler divergence (Kullback, 1959), which is a measure of distance between distributions - see Appendix B for further details.

     In simple words, when maximising the objective in Eq. 8 with respect to $\theta$ and $\phi$, we demand the encoded distribution to match the prior $p(\boldsymbol{z})$ as close as possible, while requiring that the decoded data points resemble the input data. For a full

derivation of the ELBO and further details about VAEs see e.g. Kingma and Welling (2013); Doersch (2016); Odaibo (2019), and references therein.



VAEs can be used both as a compression algorithm and a generative method. Since we want to map source coordinates to seismograms, we choose to employ a supervised version of VAEs, called Conditional Variational Autoencoders (CVAEs, Sohn et al., 2015), which proposes to maximise this slightly altered loss function:


$$\mathcal{L}(\theta, \phi; \boldsymbol{x}, \boldsymbol{c}) = \mathbb{E}_{\boldsymbol{z}} \left[ \log p_\phi(\boldsymbol{x}|\boldsymbol{z}, \boldsymbol{c}) \right] - \mathrm{D}_{\mathrm{KL}} \left( q_\theta(\boldsymbol{z}|\boldsymbol{x}, \boldsymbol{c}) || p(\boldsymbol{z}|\boldsymbol{c}) \right) ,$$ (9)

where $\boldsymbol{c}$ refers to the coordinates associated to the seismograms $\boldsymbol{x}$, and the expectation value is over $\boldsymbol{z} \sim q_\theta(\boldsymbol{z}|\boldsymbol{x}, \boldsymbol{c})$.

In our analysis, we set a latent space size of $z_{\mathrm{dim}} = 5$. Moreover, we choose the encoding $q_\theta(\boldsymbol{z}|\boldsymbol{x}, \boldsymbol{c})$ to be a multivariate normal distribution with mean given by the encoder and covariance matrix $\Sigma = 0.001^2 \mathbf{I}_{z_{\mathrm{dim}}}$. We choose a multivariate normal distribution with zero mean and the same covariance matrix $\Sigma$ as our prior $p(\boldsymbol{z}|\boldsymbol{c})$, and we employ a deterministic $p_\phi(\boldsymbol{x}|\boldsymbol{z}, \boldsymbol{c})$

as our decoding distribution. We estimate the expectation value in Eq. 9 using a Monte Carlo approximation, and we calculate the KL divergence in closed form as both $q_\theta(\boldsymbol{z}|\boldsymbol{x}, \boldsymbol{c})$ and $p(\boldsymbol{z}|\boldsymbol{c})$ are multivariate normal distributions; see Appendix B for the full derivation. The choice of $\Sigma$ is made in order to limit the spread of points in latent space, such that we can approximate the desired deterministic mapping with the probabilistic model offered by the CVAE. Once trained, we can feed a set of coordinates $\boldsymbol{c}$ and a vector $\boldsymbol{z} \sim p(\boldsymbol{z}|\boldsymbol{c})$ to the decoder to obtain a seismogram; with our setup, we verified that using a sample $\boldsymbol{z} \sim p(\boldsymbol{z}|\boldsymbol{c})$

or the mean value $\boldsymbol{z} = \boldsymbol{0}$ has no significant impact on the final performance of the model. Image (F) in Fig. 3 summarises the emulation framework making use of the CVAE trained decoder, while the architecture of the full CVAE, with hyperparameters optimised for application to the test case of Sec. 3, is shown in detail in Fig. 5.

### 2.2.7 Wasserstein Generative Adversarial Networks - Gradient Penalty ('WGAN-GP')

One of the main lines of research in generative models is based on Generative Adversarial Networks (GANs, Goodfellow et al.,
2014). In this framework, two neural networks, called generator (G) and discriminator (D), are trained simultaneously with two different goals. While G maps noise to candidate fake samples which resemble the training data to fool the discriminator, D is trained to distinguish between these fake samples and the real data points.

More formally, we can define a value function as:

$$V(D, G) = \mathbb{E}_{\boldsymbol{x}} \left[ \log D(\boldsymbol{x}) \right] + \mathbb{E}_{\boldsymbol{z}} \left[ \log \left( 1 - D(G(\boldsymbol{z})) \right) \right] ,$$ (10)

where $\boldsymbol{x}$ refers to the training data sampled from the data distribution $p_{\mathrm{data}}(\boldsymbol{x})$, and $\boldsymbol{z}$ to a noise variable sampled from some prior $p(\boldsymbol{z})$. The discriminator is thus trained to maximise $V(D)$, while the generator aims at minimising $V(D)$; the two networks play a minimax game until a Nash equilibrium is (hopefully) reached (Goodfellow et al., 2014; Che et al., 2016; Oliehoek et al., 2018).

In practice, despite generating sharp images, GANs have proved to be quite unstable at training time. Moreover, it has been
shown how vanilla GANs are prone to *mode collapse*, where the generator just focuses on a few modes of the data distribution and yields new samples with low diversity (see e.g. Metz et al., 2016; Che et al., 2016).

Many alternatives to vanilla GANs have been proposed to address these issues. We focus here on Wasserstein GANs - Gradient Penalty (WGAN-GP; Arjovsky et al., 2017; Gulrajani et al., 2017). To avoid confusion, we stress that the acronym





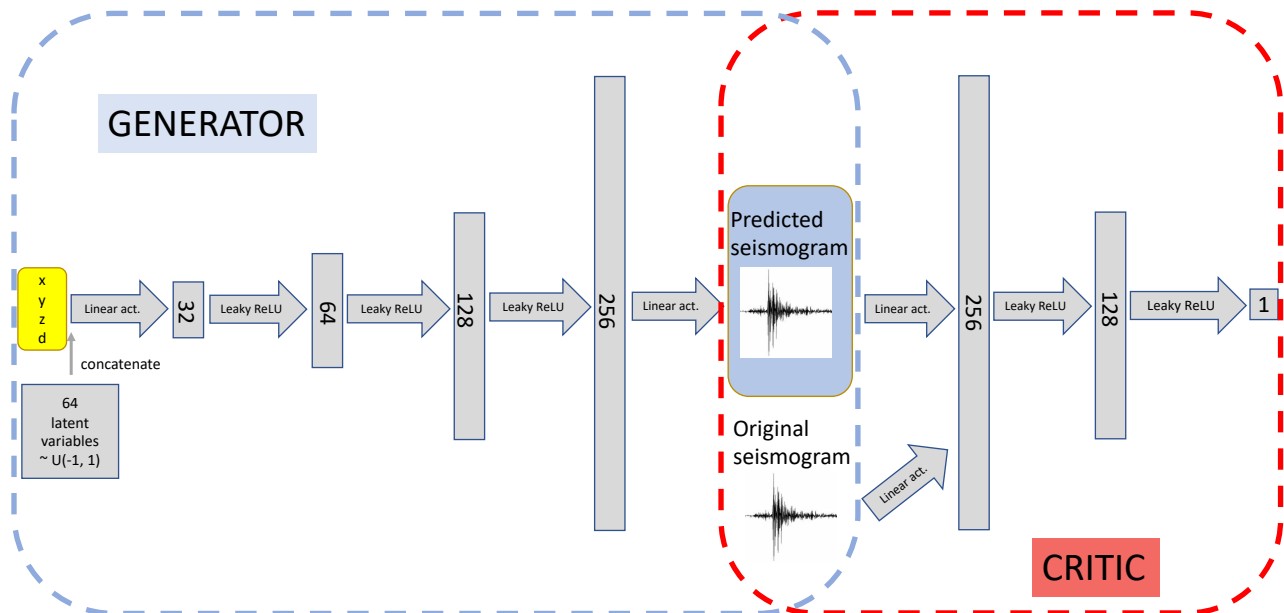

**Figure 6.** Schematic of the Wasserstein Generative Adversarial Network - Gradient Penalty described in Sec. 2.2.7. The network is composed of a generator part (in *blue*) and a critic part (in *red*). Once the full network has been trained, the generator can be removed to be used as a generative model.

'GP' is used to indicate a Gaussian Process throughout the paper, while it refers to the 'Gradient Penalty' variant of the WGAN algorithm only when quoted as 'WGAN-GP'. In short, we still consider two networks, a generator G and a critic C, which are trained to minimise the following objective:

$$\min_{G} \max_{C} \mathbb{E}_{\boldsymbol{x},\boldsymbol{c}}\left[\log C(\boldsymbol{x},\boldsymbol{c})\right] - \mathbb{E}_{\boldsymbol{z},\boldsymbol{c}}\left[\log\left(1 - C(G(\boldsymbol{z},\boldsymbol{c}))\right)\right] + -\lambda\mathbb{E}_{\hat{\boldsymbol{x}},\boldsymbol{c}}\left[\left(||\nabla_{\hat{\boldsymbol{x}}}C(\hat{\boldsymbol{x}},\boldsymbol{c})||_2 - 1\right)^2\right], \tag{11}$$

where $\boldsymbol{c}$ refers to the coordinates, $\hat{\boldsymbol{x}}$ is a linear combination of the real and generated data, $\lambda \geq 0$ is a penalty coefficient for the regularisation term, and $||\nabla_{\hat{\boldsymbol{x}}}||_2$ refers to the $L^2$ norm of the critic's gradient with respect to $\hat{\boldsymbol{x}}$. See Appendix C for further details. Image (G) in Fig. 3 summarises the emulation framework making use of the generator of the WGAN-GP, whose full architecture is described in detail in Fig. 6.

In our experiments, we chose $\lambda = 10$, and trained the critic $n_{\mathrm{crit}} = 100$ times for every generator weight update. Both our generator and discriminator are made of fully-connected layers with various numbers of hidden neurons. We set the dimension of the latent $z_{\mathrm{dim}} = 64$, and $p(\boldsymbol{z}) \sim U(-1,1)$. Note that the choice of how to include the conditional information in the architecture is not unique, and we experimented with different combinations without significant differences. Once the algorithm has been trained, a new seismogram is obtained by feeding the generator with a latent vector and a set of coordinates.





Finally, note that, in this case only, we standardised the data $x$ after the rescaling described in Sec. 2.1. We calculated the mean $\mu$ and the standard deviation $\sigma$ over all seismograms $x$, and trained our model on $x' = \frac{x-\mu}{\sigma}$.

## 2.3 Training, validation and testing procedure

We describe here the methodology followed to train our models and test their accuracy. We remark that the training and testing of any machine learning algorithm should be performed on a case-by-case basis, in order to match the accuracy requirements dictated by the specific problem considered (in this case, the emulation of seismic traces given a certain velocity model). For concreteness, we present here the details of training and testing our models for application to the test case in Sec. 3.

   All our models are trained on the same 2000 simulated events used in D18. For optimisation and testing purposes, we divide

the remaining 2000 samples (from the pool of 4000 events generated in total by D18) into a validation set and a testing set of 1000 events each. Differently from D18, in this paper we use a *validation* set to tune the hyperparameters of our deep learning models. To provide an unbiased estimate of the performance of the final tuned models, we quote our definitive results evaluating the accuracy of each model on the *testing* set, which is never 'seen' by the model at any point in the training or optimisation procedures.

Similar to D18, our accuracy performance is quantified in terms of the $R_{2D}$ coefficient, a standard statistics commonly used in time series analysis to quantify the correlation between two signals. Given a batch of the true seismograms $\mathbf{G}$ and the corresponding emulated ones $\mathbf{P}$, the $R_{2D}$ coefficient is defined as

$$R_{2D} = \frac{\sum_i \sum_j \left(G_{ij} - \bar{G}\right)\left(P_{ij} - \bar{P}\right)}{\sqrt{\left(\sum_i \sum_j \left(G_{ij} - \bar{G}\right)^2\right)\left(\sum_i \sum_j \left(P_{ij} - \bar{P}\right)^2\right)}} \ , \tag{12}$$

$$\bar{G} = \frac{1}{N_s}\frac{1}{N_t}\sum_i \sum_j G_{ij} \ , \quad \bar{P} = \frac{1}{N_s}\frac{1}{N_t}\sum_i \sum_j P_{ij} \ ,$$

where $\bar{G}$ and $\bar{P}$ are the mean over all $i = 1, \ldots, N_s$ samples and $j = 1, \ldots, N_t$ time components of the ground truth $G_{ij}$ and predicted seismograms $P_{ij}$. Given its normalisation, the $R_{2D}$ coefficient ranges between values of $-1$, denoting perfect anti-correlation, to $+1$, indicating perfect correlation; a vanishing correlation coefficient denotes absence of correlation.

   When training our NNs, all implemented in TENSORFLOW(Abadi et al., 2015), we monitor the value of the validation loss to choose the total number of epochs, waiting 100 epochs after the loss stopped decreasing and restoring the model with the

lowest validation loss value. In other words, we early-stop (Yao et al., 2007) based on the validation loss with patience = 100 epochs. Moreover, we optimise our algorithms calculating the final $R_{2D}$ coefficient, as defined in Eq. 12, over different combinations of the hyperparameters, choosing the values that yield the highest $R_{2D}$. The optimisation procedure is performed using the adaptive learning rate method Adam (Kingma and Ba, 2014), with default parameters. The optimisation of the network hyperparameters is entirely performed on the validation set; the testing set is left unseen by the networks until the very

last stage of the analysis, when it is used to calculate the results quoted in Tab. 1.





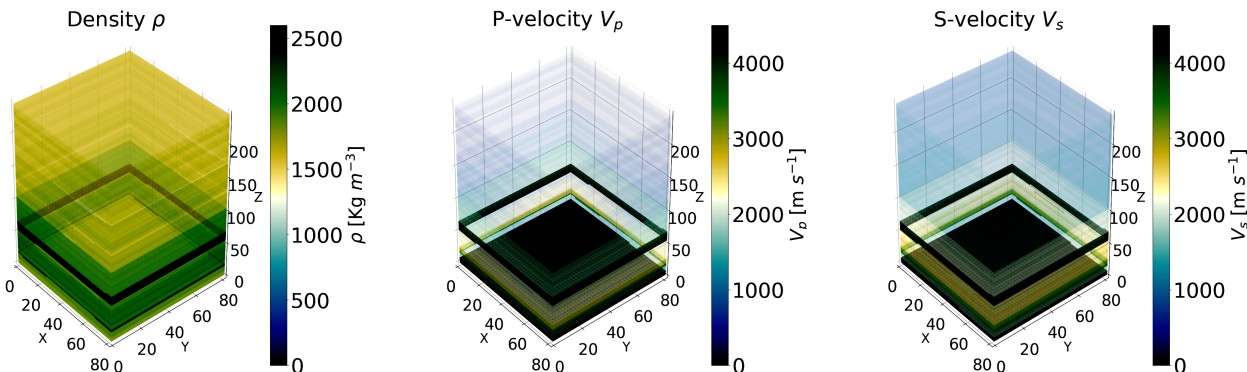

**Figure 7.** Density, P-wave velocity and S-wave velocity models of the simulated domain used in this work. The models are specified as 3D grids of voxels. Notice the layered structure, with variation along the vertical dimension much more pronounced than across horizontal planes.

## 3 Application

Since one of our goals is to compare our new emulation methods with the one previously developed in D18, we train and test them on the same geophysical scenario considered there. To train and test our algorithms we use the same microseismic traces that were forward modelled in D18 for training and testing purposes.

### 320 3.1 Simulation setup

We briefly recap here the characteristics of the simulated geophysical domain and microseismic traces, referring to D18 for further details. We consider a geophysical framework where we record seismic traces in a marine environment. Sensors are placed at the seabed to record both pressure and three-component particle velocity of the propagating medium. As was the case in D18, we assume that our recorded seismic traces are generated by explosive isotropic sources. For isotropic sources,

considering only the pressure wave and ignoring the particle velocity is sufficient to determine the location of the event in the studied domain, as shown in D18. We consider the seismic traces to be noiseless when building the emulator, while some noise is added to the simulated recorded seismogram when inferring the coordinates' posterior distribution, as we will show in Sec. 3.2.

Forward simulations of seismic traces are obtained by solving the elastic wave equation given a 3D heterogenous velocity

and density model for the propagating medium, shown in Fig. 7. The model specifies values of the propagation velocities for P- and S-waves $(V_p, V_s)$ as well as the density $\rho$ of the propagating medium, discretised on a 3D grid of voxels. The solution of the elastic wave equation is a computationally challenging task, which can be accelerated using GPUs (Das et al., 2017). This is implemented in the software K-WAVE (Treeby et al., 2014), a pseudospectral method employed by D18 to generate



| Model | $R_{2D}$ | Training time (s) | Evaluation time (ms) | Size (MB) |
|---|---|---|---|---|
| NN direct (A) | $0.9500 \pm 0.0006$ | $270 \pm 12$ | $9.9 \pm 0.6$ | 2.32 |
| PCA + NN (B) | $0.9443 \pm 0.0006$ | $180 \pm 1$ | $8.6 \pm 0.2$ | 0.71 |
| PCA + GP (C) | $0.9433 \pm 0.0006$ | $1463 \pm 27$ | $97.9 \pm 2.6$ | 1.52 |
| AE + NN (D) | $0.9496 \pm 0.0021$ | $228 \pm 12$ | $9.3 \pm 1.0$ | 4.46 |
| AE + GP (E) | $0.9472 \pm 0.0029$ | $488 \pm 23$ | $25.4 \pm 1.4$ | 4.59 |
| CVAE (F) | $0.9477 \pm 0.0005$ | $302 \pm 7$ | $9.3 \pm 0.5$ | 4.37 |
| WGAN-GP (G) | $0.9214 \pm 0.0048$ | $1069 \pm 59$ | $9.9 \pm 0.3$ | 4.07 |
| D18 | $\sim 0.89$ | 29232 | $621.0 \pm 8.8$ | 5.12 |

**Table 1.** 2D correlation coefficient $R_{2D}$ (as defined in Eq. 12), training time, single likelihood evaluation time and total size of the model for all our models and the model of Das et al. (2018, D18). The capital letter in round brackets refers to the schematic in Fig. 3. Note that training time refers to the total time to preprocess, train and postprocess data. All of our experiments were run on an Intel® Core™ i7-8750H CPU @ 2.20GHz, which can be found on an average-performing laptop. Results for D18 are taken from Tab. 2 and Fig. 14 in D18, and have been run in parallel on an HPC cluster, with the only exception of the likelihood evaluation time, which we performed on our machine. The reported values of $R_{2D}$ and times are the mean and standard deviation of 3 runs. All of our proposed models perform better than the one shown in D18, while taking considerably less time and requiring less disk space and hardware performance.

their training and testing samples, which we also use in our analysis. The GPU software allows for the computation of the acoustic pressure measured at specified receiver locations. 4000 microseismic traces are generated in total with a NVIDIA P100 GPU, given their random locations within a predefined domain and a specified form for their moment tensor. The value of the diagonal components of the moment tensor is set equal to 1 MPa, following Collettini and Barchi (2002). The coordinates $(x, y, z)$ of the simulated sources are sampled using Latin Hypercube Sampling on a 3D grid of $81 \times 81 \times 301$ gridpoints, corresponding to a real geological model (the same used in D18) of dimensions 1 km $\times$ 1 km $\times$ 3 km. The temporal sampling

interval for the solution of the elastic wave equation is 0.8 ms, which ensures stability of the numerical solver. The synthetic traces have a total duration of 2 s each. After generation, all seismograms are downsampled to a sampling interval of 4 ms to reduce computational storage. This way, each seismic trace is ultimately a time series composed of $N_t = 501$ time components. Note that, as explained previously, similarly to D18, we augment each of our $(x, y, z)$ coordinate set with their distance from the receiver $d = \sqrt{x^2 + y^2 + z^2}$, as we noted that this helps the training of the generative models.

## 3.2 Comparison with Das et al. (2018)

In this Section we summarise our main findings. We start in Sec. 3.2.1 by describing the accuracy performance of all our new methods, and compare them with that achieved by D18. In Sec. 3.2.2 we then move to our inference results, describing how we simulated a microseismic measurement and used our generative models to accelerate Bayesian inference of the event coordinates, again comparing against the results obtained applying the method described in D18.





**Figure 8.** Comparison of the reconstruction accuracy of different emulation methods on three random seismograms from the testing set (in *black, dashed* line), whose coordinates are reported on top of each panel. The seismograms record the vertical component of motion at the receiver placed on the point with coordinates $(0.5\mathrm{km}, 0.5\mathrm{km}, 2.43\mathrm{km})$. The horizontal axis is zoomed around the location of the P-wave peak. In addition to the D18 method (in *blue*), we show the performance of the methods achieving lowest and highest accuracy as reported in Tab. 1: the 'WGAN-GP' model (*pink*) and the 'NN direct' model (*red*), respectively.





### 3.2.1 Performance of the generative models


In Tab. 1 we report summary statistics for the performance of our generative models. Our goal is to critically compare the different methods, highlighting their strengths and weaknesses, so that the reader can decide to adopt the one that fits best their primary interests and available resources. To perform this comparison, similarly to D18, we consider an experimental setup with only one central receiver at planar coordinates ($x = 0.5 \text{ km}, y = 0.5 \text{ km}$) in the detection plane $z = 2.43 \text{ km}$ (see Fig. 9).

In the following paragraphs we will then consider a more complicated geometrical setup for the detection of microseismic events.

Considerations of *accuracy* in terms of reconstructed seismograms are important for applications to posterior inference analysis, to avoid biases and/or misestimates of the uncertainty associated to the inferred parameters. In our case achieving higher accuracy is crucial to guarantee unbiased and accurate estimation of the microseismic source location. For this reason,

in Tab. 1 we cite the $R_{2D}$ statistic defined in Eq. 12 as a means to quantify the accuracy of our methods, similarly to what was done in D18. The $R_{2D}$ coefficient is evaluated on the testing set, after training and validation of each method, according to the procedure described in Sec. 2.3.

All of our new methods provide a $R_{2D}$ statistic higher than the one reported by D18 on their testing set. We note that in D18 the testing set was composed of 2000 events, whereas here we split those 2000 events in a validation and testing set of

1000 samples each. However, we checked that all of our numerical conclusions are unchanged considering a larger testing set composed of the same 2000 events used by D18. We also checked that training the D18 emulator (augmented with the two GPs for the ($\bar{A}, \bar{t}$) coefficients) on the seismograms preprocessed following Sec. 2.1 leads to values of $R_{2D}$ worse than that obtained applying the D18 method without preprocessing. Hence, for our comparison we decided to leave the D18 method unchanged from its original version, i.e., without performing the preprocessing of Sec. 2.1 prior to training.

The 'NN direct' model, described in Sec. 2.2.1, provides the highest $R_{2D}$ value among our proposed methods. This is due to the combination of two factors: the relatively simple structure of the seismograms, given the isotropic nature of their moment tensor, and the preprocessing operated on the training seismograms. On the one hand, isotropic sources are characterised by strong and very localised P-wave peaks, which clearly dominate over the rest of the signal. This simplifies the form of the signal with respect to e.g. pure compensated linear vector dipole (CLVD) and double-couple (DC) events, characterised by

more complicated signal structure (Vavryčuk, 2015; Das et al., 2017). On the other hand, even with the relatively simple structure of the isotropic seismograms, the training of a NN to map coordinates to seismic traces is extremely challenging due to the reduced number of training samples available. It is for this reason that we operated the preprocessing on the training seismograms described in Sec. 2.1. This has the advantage of extracting information regarding the source-sensor distance, encoded mainly in the location and amplitude of the P-wave peak of each seismic trace. By isolating these features into the

parameters ($\bar{A}, \bar{t}$), we simplify the task for our NN or any other method learning the mapping between source coordinates and seismograms. This approach relies on being able to train methods that learn efficiently the mapping between coordinates and ($\bar{A}, \bar{t}$) coefficients. Fortunately, this mapping is not too complicated, depending mainly on the distance of the source from the





sensor, and this is quite simple to learn for the GPs described in Sec. 2.1 which, as we verified experimentally, show higher accuracy than NNs in learning the $(\bar{\boldsymbol{A}}, \bar{\boldsymbol{t}})$ coefficients.

Fig. 8 shows the reconstruction accuracy of three models among the ones considered in Tab. 1, namely the D18 method and the two models proposed in this paper which yield lowest and highest $R_{2D}$ coefficient (the 'WGAN-GP' and 'NN direct' methods, respectively). We evaluate the predictions of these three models for three random coordinates among those of the testing set, and check how the predictions compare with the original seismograms. We notice how in some cases the D18 method fails to produce accurate predictions (as in the case of the seismogram shown in the second and third column in Fig. 8).

The 'WGAN-GP' and 'NN direct' methods, instead, manage to yield more accurate predictions in these cases, in particular regarding the location and amplitude of the P-wave peak, crucial for localisation purposes. From the Figure we can appreciate how even the 'WGAN-GP' method, whose accuracy is the worst among the methods proposed in this paper (cf. Tab. 1), reconstructs the seismograms in the second and third column better than the D18 method.

  *Speed* considerations are also important when evaluating the performance of the models. In general, applications of deep
learning to Bayesian analysis may often be possible only making use of High Performance Computing (HPC) infrastructures. If these are not available, applications to real parameter estimation frameworks may be fatally compromised. Therefore, it is important to notice that all our proposed models can be efficiently run on a simple laptop, without the need of any HPC platform. If HPC infrastructures are available, they would speed up our models even further. In particular, running all generative models on GPUs would lead to a speed-up of at least an order of magnitude (Wang et al., 2019).

Importantly, however, even without this HPC acceleration we find that all our models are $\sim$ 1-2 orders of magnitude faster to train *and* to evaluate than the method described in D18. We stress here that the advantage of our models in terms of speed relies not only in requiring considerably less time to train, but also, and arguably more importantly, in predicting a seismogram much faster than with the D18 method. This point is essential for applications to parameter inference, e.g. through MCMC techniques, where a forward model needs to be computed at each likelihood evaluation. A single-evaluation time for our
models is reduced of up to 2 orders of magnitude with respect to that of D18, which in turns means that Bayesian inference of microseismic sources will be similarly faster (see Sec. 3.2.2). Our emulators run on a common laptop CPU provide a $\mathcal{O}(10^5)$ speed-up compared to direct simulation of the seismic trace with a pseudo-spectral method run on a GPU. The training time required by each method is also significantly lower than in D18. We note that this last property makes the training of our models much less demanding in terms of computational resources. We also note that the creation of a training dataset, with a
few thousands seismograms generated by solution of the elastic wave equation, is a computational overhead cost that we share with the analysis of D18, and therefore its generation time is not reported here for any of the methods in Tab. 1, including D18 (see Sec. 4 for a discussion on how to reduce this overhead simulation time in future work).

  Among our proposed methods, the fastest to evaluate is the 'PCA+NN' method described in Sec. 2.2.2. This was expected, as this model is composed of a relatively small NN and a reconstruction through the predicted PCA coefficients. Both operations
essentially boil down to matrix multiplications, which can be executed with highly optimised software libraries. We also notice that the methods requiring GP predictions ('PCA+GP' and 'AE+GP' in Tab. 1) are the ones that perform worse in terms of evaluation and training speed. Again, this was expected as it is due to the nature of GP regression itself. Contrary to NNs,




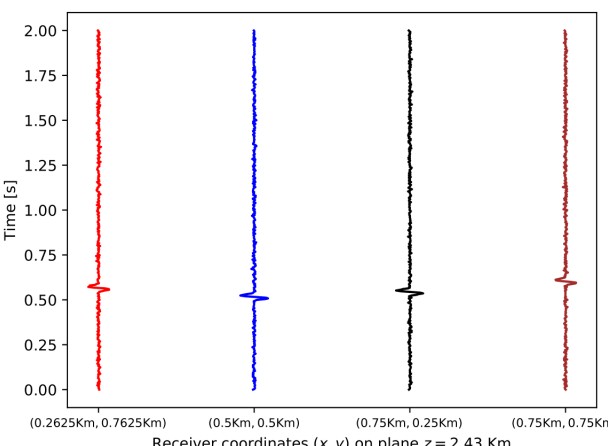
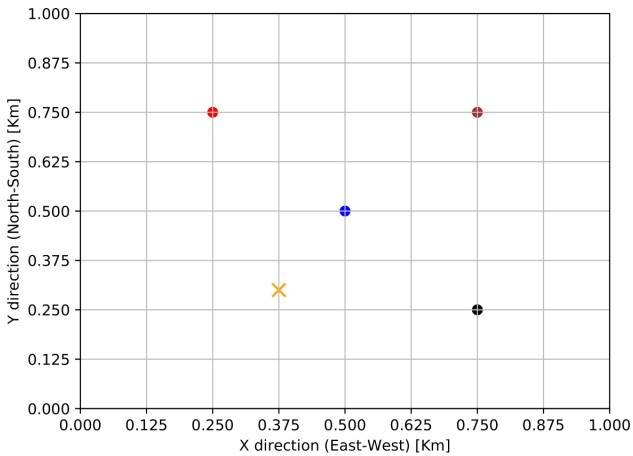

**Figure 9.** *Left panel:* simulated noisy seismic traces recorded by the sensors at the seabed, with configuration shown in the *right* panel. These recorded seismograms represent the data vector for our simulated posterior distribution inference. *Right panel:* simulated receiver geometry in the detection plane $z = 2.43$ km. The dots indicate the sensor locations, with colours matching those of the recorded seismic traces in the *left* panel. The central receiver with coordinates $(0.5\text{km}, 0.5\text{km}, 2.43\text{km})$ is the one that we consider (similarly to D18) when we quantify the performance of our trained generative models in Table 1. We then include the other receivers when we demonstrate the effectiveness of our models in carrying out Bayesian inference of the coordinates of a simulated seismic event with coordinates $(0.375\text{km}, 0.3\text{km}, 1.57\text{km})$, whose projection on the detection plane is marked with an *orange* cross.

GPs are non-parametric methods that need to take into account the entire training dataset each time they make a prediction. At inference time they need to keep in memory the whole training set and the computational cost of predictions scales (cubically)

with the number of training samples (Liu et al., 2018). This affects also the D18 method, in an even more exacerbated form since the number of GPs involved in that method is higher.

     Related to the difference between GP and NN regression are the *storage size* requirements of the different methods. Models employing NNs are less demanding than GPs in terms of memory requirements, mainly because they do not need to keep memory of the training data. Within NN architectures, the simpler ones are, intuitively, the lightest to store. 'PCA+NN' is

again, the best performing method in this regard, winning in particular over 'AE+NN' since the latter requires the storage of weights and biases for two NNs.

### 3.2.2   Inference results

Now that we have quantified the performance of our generative models we want to apply them to the Bayesian inference of a microseismic event location. To this purpose, we simulate the detection of a microseismic event and wish to infer the posterior

distribution of its coordinates. The posterior distribution of a set of parameters $\theta$ for a given model or hypothesis $H$ and a data





| Coord. | Prior range [km] | Ground Truth [km] | D18 [km] | NN direct [km] | EDT [km] |
|:------:|:----------------:|:-----------------:|:--------:|:--------------:|:--------:|
| $x$ | $[0, 1]$ | 0.3750 | $0.2175^{+0.5325}_{-0.12375}$ | $0.3770^{+0.0025}_{-0.0025}$ | $0.383^{+0.022}_{-0.022}$ |
| $y$ | $[0, 1]$ | 0.3 | $0.2^{+0.525}_{-0.125}$ | $0.305^{+0.0026}_{-0.0026}$ | $0.32^{+0.022}_{-0.022}$ |
| $z$ | $[0, 2.43]$ | 1.57 | $0.33^{+0.64}_{-0.28}$ | $1.57^{+0.0012}_{-0.0012}$ | $1.57^{+0.015}_{-0.015}$ |

**Table 2.** Prior range and mean and marginalised 68 percent credibility intervals on the coordinates $(x, y, z)$, for the D18 method, our proposed 'NN direct' model described in Sec. 2.2.1, and the EDT time arrival inversion described in Sec. 3.3.

set $\boldsymbol{D}$ is given by Bayes' theorem (e.g. MacKay, 2003)

$$\Pr(\boldsymbol{\theta}|\boldsymbol{D}, H) = \frac{\Pr(\boldsymbol{D}|\boldsymbol{\theta}, H)\Pr(\boldsymbol{\theta}|H)}{\Pr(\boldsymbol{D}|H)}. \tag{13}$$

The *posterior* distribution $\Pr(\boldsymbol{\theta}|\boldsymbol{D}, H)$ is the product of the *likelihood* function $\Pr(\boldsymbol{D}|\boldsymbol{\theta}, H)$ and the *prior* distribution $\Pr(\boldsymbol{\theta}|H)$ on the parameters, normalised by the *evidence* $\Pr(\boldsymbol{D}|H)$, usually ignored in parameter estimation problems since it is inde-

pendent of the parameters $\boldsymbol{\theta}$. In this work we employ the algorithm MULTINEST (Feroz et al., 2009) for multi-modal nested sampling (Skilling, 2006), as implemented in the software PYMULTINEST (Buchner et al., 2014), to sample the posterior distribution of our model parameters (i.e., the source coordinates).

We simulate the observation of a microseismic isotropic event, by generating a noiseless trace given specified coordinates $(x, y, z) = (0.375\,\text{km}, 0.3\,\text{km}, 1.57\,\text{km})$. Our goal is to derive posterior distribution contours on the coordinates $x, y, z$, which

represent our parameters. Following D18, we add random Gaussian noise to each component of the noiseless trace, with standard deviation $\sigma = 250$ in the same arbitrary units as the seismograms' amplitude. The resulting seismic trace, measured at multiple receivers, is shown in the left panel of Fig. 9. Similarly to D18, we assume a Gaussian likelihood. We stress here that the particular shape considered for the noise modelling and the likelihood function are not restrictive: our methodologies are easily applicable to more complicated noise models or likelihood forms, while we chose to use the same investigated by D18

for a direct and fair comparison.

Instead of repeating the analysis for each proposed generative model, we decide to use the one that has been shown in Tab. 1 to achieve greater accuracy, i.e., the direct neural network mapping between coordinates and seismograms, described in Sec. 2.2.1. We simulate an experimental setup with multiple receivers on the detection plane $z = 2.43$ km, shown in the right panel of Fig. 9. D18 reported a maximum likelihood calculation for up to 23 receivers placed on the same plane. Here,

our aim is to test the performance of our models at inference time, while we do not wish to carry out a detailed analysis for optimisation of the receivers geometry. In particular, we wish to compare the D18 emulator with ours at inference time. Thus, we do not consider all 23 receivers considered by D18. While we find that considering only one receiver is obviously not enough to achieve significant constraints on the coordinates, after experimenting with different configurations and number of receivers we find that considering four receivers, placed in the upper diagonal part of the detection plane as shown in

Fig. 9, leads to significant constraints on the event coordinates. This is true if we consider our 'NN direct' method, whose accuracy is higher than the one of D18. Indeed, repeating the inference analysis with the D18 emulator, given the same receiver configuration, leads to constraints $\sim 3$ times less tight on the coordinates and at a considerable increase in computation time:


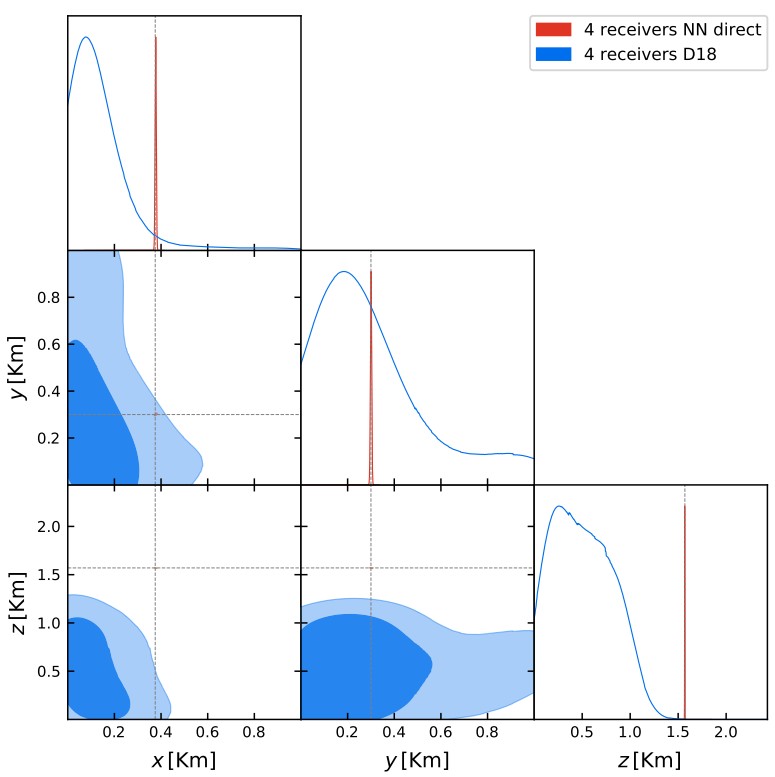

**Figure 10.** Comparison of the marginalised 68 and 95 per cent credibility contours obtained with the D18 method (in *blue*) and our proposed 'NN direct' generative model (in *red*) described in Sec. 2.2.1, considering a seismic trace measured by the four receivers shown in Fig. 9. The *black* dashed lines indicate the source's true coordinates at $(x, y, z) = (0.375\text{km}, 0.3\text{km}, 1.57\text{km})$.

66h of computation using 24 Central Processing Units (CPUs), compared to the 1.7h required by the 'NN direct' on a single CPU. The fact that such tighter constraints can be achieved with our emulator, even if making use of the information coming
from only four receivers, is due to the increased accuracy of our method, evident from the $R_{2D}$ values reported in Tab. 1.

     Fig. 10 shows the posterior contour plots for the four receiver configuration described above, obtained with our 'NN direct' generative model and the emulator of D18. The numerical results are summarised in Tab. 2, reporting the prior ranges and mean and marginalised 68 percent credibility interval on the coordinates. We notice that the $x$ and $y$ coordinates are less constrained than the $z$ coordinate. This is due to the layered structure of the density and velocity model (cf. Fig. 7), with much
more variability along $z$ than along the horizontal directions. A full comparison between the D18 and 'NN direct' methods would require to perform the inference process using data from all 23 receivers. However, we found that implementing the D18 method with all 23 receivers involves significant computational complication, even when making use of highly parallelised HPC implementations. We remark that the D18 method fails with few detectors and is computationally expensive with many,





while the 'NN direct' method proposed in this paper works well with just 4 detectors and can be expected to work very well,
and at lower cost, with many.

### 3.3  Comparison with arrival time - based nonlinear location

In order to check the accuracy of our emulators, we perform a comparison of the inference results that we obtain with our surro-
gate model with the results obtained from a non-linear probabilistic location method. The widely used algorithm NONLINLOC
(Lomax et al., 2000) allows the determination of source locations based on arrival time data using a probabilistic approach. It
implements the Equal-Differential-Time formulation (EDT; Zhou, 1994; Font et al., 2004; Lomax, 2005), which uses relative
arrival time between stations to remove the origin time from the parameter space. In this work we consider the origin time to
be at $t = 0$ and we do not include it in our parameter space, so strictly we do not need the EDT formulation. However, we
stick to it to produce results easily comparable with standard procedures. In the notation of Lomax et al. (2009) and the EDT
formulation, the likelihood of the observed arrival times takes the form

$$
\qquad \left[ \sum_{a,b} \frac{1}{\sqrt{\sigma_a^2 + \sigma_b^2}} \cdot \exp\left( -\frac{\left( [T_a^0 - T_b^0] - [TT_a^0 - TT_b^0] \right)^2}{\sigma_a^2 + \sigma_b^2} \right) \right]^N , \qquad (14)
$$

where the indices $a, b$ run over the receivers, $T_a^0$ and $T_b^0$ are observed arrival times, $TT_a^0$ and $TT_b^0$ are theoretical estimates for
the travel times, the uncertainties $\sigma_a, \sigma_b$ combine errors in the arrival time theoretical calculations and observations, and $N$ is
the total number of observation ($N = 1$ in our test case). The NONLINLOC software uses this likelihood function to sample
the posterior distribution of the model parameters (i.e., the coordinates of the recorded seismic event) given specified priors,
which we take here to be uniform in the same ranges considered in 3.2.2. While we use this likelihood function for comparison
with NONLINLOC, we do not sample the posterior distribution using the sampling algorithms implemented in the software
package. Instead, we build an independent implementation that uses PYMULTINEST for sampling the posterior distribution, for
comparison with section 3.2.2, noting that the inference results are independent of the algorithm used for posterior sampling.

The theoretical computations of the arrival times given specified coordinates $(x, y, z)$ in the simulated 3D domain are ob-
tained with the algorithm PYKONAL (White et al., 2020), which implements a Fast Marching Method (Sethian, 1996) to solve
the eikonal equation in Cartesian or spherical coordinates in 2 or 3 dimensions.

Our goal is to verify that our methodology provides estimates of the posterior probability distribution for the event location
which is in good agreement with that obtained from the NONLINLOC method. To obtain this comparison, we consider the
same 4 receivers shown in the right hand panel of Fig. 9. The simulated observation is also the same one considered in the
comparison with the D18 method and shown in the left hand panel of Fig. 9. In particular, the noise properties of the observed
waveform for our method are the same considered in Sec. 3.2.2, i.e., Gaussian noise with standard deviation $\sigma = 250$ in the
arbitrary units of pressure used in this work. For the arrival time observation of the NONLINLOC method, we consider an error
on the manual picking of the arrival time of $0.005$ s, following Smith (2019).

The inference results for the posterior distribution of the coordinates are reported in Tab. 2 and shown in Fig. 11, super-
imposed with our constraints obtained with the 'NN direct' method. Clearly, with the latter we obtain constraints in good



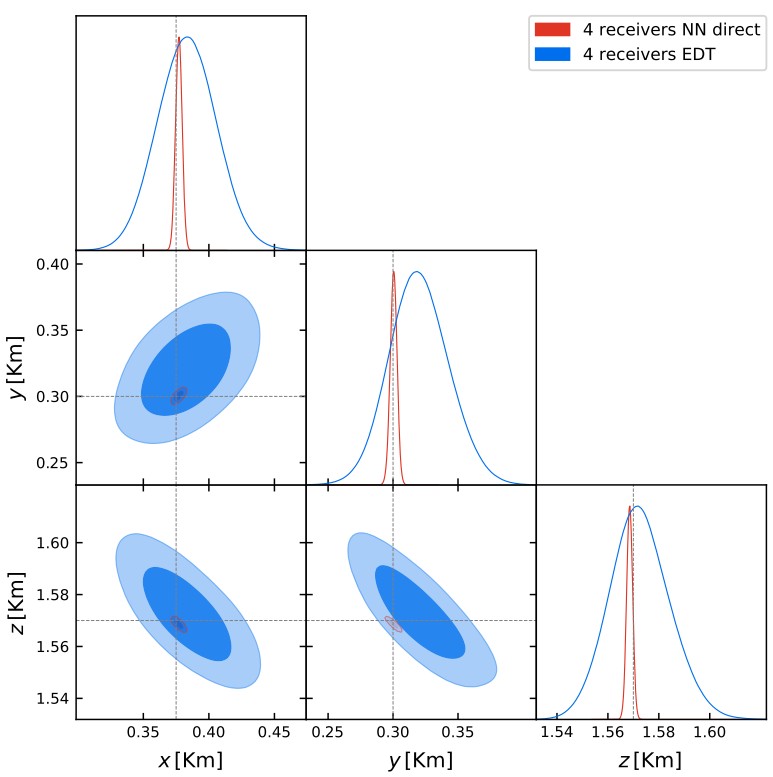

**Figure 11.** Comparison of the marginalised 68 and 95 per cent credibility contours obtained with the EDT nonlinear location method (in *blue*) and our proposed 'NN direct' generative model (in *red*) described in Sec. 2.2.1, considering a seismic trace measured by the four receivers shown in Fig. 9. The *black* dashed lines indicate the source's true coordinates at $(x, y, z) = (0.375\,\text{km}, 0.3\,\text{km}, 1.57\,\text{km})$.

agreement with the NONLINLOC method, while being tighter, as expected, since we are using the full waveform information rather than the arrival time only. However, we also notice that this comparison is strongly dependent on the uncertainty associated with the estimated arrival time in the NONLINLOC method, and the fact that we neglect possible errors associated with the theoretical predictions for the ray-traced estimates of the travel times from PYKONAL. For this reason, we emphasise

here that this comparison should not be regarded as a way to show that our method certainly provides tighter constraints on the source coordinates, although this could be expected since we are considering the full waveform information as opposed to merely inverting the arrival times. We leave a proper comparison of the uncertainties associated with both approaches to future work, but we can already notice that both methods give results in good agreement.

## 4   Conclusions

In this paper we developed new generative models to accelerate Bayesian inference of microseismic event locations. Our geophysical setup was similar to the one used in Das et al. (2018, D18) to train an emulator with the aim of speeding up the



source location inference process. This was achieved by replacing the computationally expensive solution of the elastic wave equation at each point in the parameter space explored by e.g. Markov Chain Monte Carlo (MCMC) techniques for posterior distribution sampling. In both D18 and this work, emulators were trained to learn the mapping between source coordinates and
seismic traces recorded by the sensor.

All models developed in this paper were trained on the same 2000 forward simulated seismograms used by D18 when training their emulator. However, our models are based on deep learning architectures and make minimal use of Gaussian Process (GP) regression, which is instead performed multiple times in the method proposed by D18. This makes all of our models faster to train and evaluate compared to the previous emulator, achieving a speed-up factor of up to $\mathcal{O}(10^2)$, as well
as reducing the storage requirements of the models. Our trained emulators are capable of producing synthetic seismic trace for a given velocity model with a speed-up factor over pseudo-spectral methods of $\mathcal{O}(10^5)$. For example, it takes $\sim 10$ ms to compute a 2-s synthetic trace for a given source model on a common laptop CPU, compared to $\sim 1$ hr using the pseudo-spectral method implemented in the software K-WAVE, run on a GPU. Crucially, this acceleration does not happen at the expense of accuracy; on the contrary, our models provide improved constraints on the source coordinates.

We showed this first by calculating the 2D correlation coefficient for the seismograms of the test set. The values obtained with all our models were higher than those obtained by D18, indicating the higher accuracy achieved. Secondly, we repeated the simulated experiment devised by D18, with sensors placed at the seabed of a 3D marine environment where our simulated sources were randomly located. We showed that using information coming from only four receivers situated on the detection plane we were able to provide accurate and tight constraints on the source coordinates, whereas the D18 method struggled to
provide any significant constraint given the same setup and would likely need additional information from more sensors to achieve comparable constraints. As a result of the speed up obtained at evaluation time, we were able to perform the inference process on a single CPU in $\sim 1.7$h, compared to $\sim 66$h of calculation over 24 CPUs required by the D18 method.

We also compared our inference results with those obtained from arrival time inversion, following the methodology implemented in the software NONLINLOC (Lomax et al., 2000). We found that our full waveform constraints are, as expected,
tighter than those obtained from arrival time inversion; however, we notice that a comparison of the constraints obtained with the two methodologies is not straightforward, as it depends on various modelling choices, most importantly the error associated to the arrival time estimate. Therefore, we argue that the comparison carried out in this paper should rather be regarded as a validation for our newly proposed method.

A complete Bayesian hierarchical model for source location has been developed in the software BAYESLOC (Myers et al.,
2007, 2009). We believe that the implementation of our emulators in this framework could benefit greatly the speed of execution of the BAYESLOC software, with potential application to e.g. the study of nuclear explosions as in Myers et al. (2007, 2009). We also notice that an arguably faster method for source location exists, which makes use of the time travel information derived from simulated waveforms (Vasco et al., 2019). This method is a variation of the grid search method of Nelson and Vidale (1990), with travel time calculations obtained from full waveform simulations instead of the solution of the eikonal equation.
We remark that this method may be preferable to ours in terms of speed, since it scales with the number of receivers in the recording network, thanks to the reciprocity relation (e.g. Chapman, 2004) used in the calculation of the travel time fields, by





placing a source in the receivers location and solving the elastodynamic equation. However, we also notice that the method of Vasco et al. (2019) ultimately makes use only of arrival time estimates, hence it is possible that our full waveform inversion may lead to tighter constraints, as seen in Sec. 3.3 (although the same caveats on the comparison performed there would equally apply in comparing with the method of Vasco et al. 2019). We also notice that the method proposed in Vasco et al. (2019) is not presented within a Bayesian framework, whereas all our emulators are. This is a key characteristic of the methods developed in our paper, in view of integration of our generative models within Bayesian frameworks for joint inversion of moment tensor components and location.

In conclusion, we provided the community with a collection of deep generative models that can accelerate very efficiently Bayesian inference of microseismic sources. The ultimate goal here would be to integrate our emulators within existing methodologies and software for joint location and moment tensor components inversion, as e.g. implemented in MTFIT (Pugh and White, 2018). We believe that the results obtained in this paper sufficiently prove the accuracy of the emulators developed, making them ready for integration within MTFIT.

The performances of our emulators in terms of accuracy are all comparable between them, and improved with respect to the D18 method. Speed considerations may therefore be invoked in the decision process for a particular method. However, we notice that our framework is valid only for microseismic events characterised by isotropic moment tensor. Considering more complicated forms of the moment tensor will likely require additional complications, first of all considering seismic traces recorded for longer time, since the signal structure will be in general more complicated. Extensions of this work to non-isotropic sources, possibly in combination with other source inversion techniques (e.g. Minson et al., 2013; Weston et al., 2014; Frietsch et al., 2019; Vasyura-Bathke et al., 2020) would then allow for an extension of the parameter space to be explored, including for example the moment tensor components for characterisation of the source mechanism. Additionally, applications to real analyses will need to implement more realistic models for the noise than the one we considered when performing Bayesian inference.

We stress, however, that since these sophistications will likely require larger training sets, the development of the generative models presented in this paper is a crucial step for future work. Once again, this is due to the fact that our models are based on deep learning architectures, reducing the use of GP regression. This is a crucial feature for extensions of this work to more complicated source mechanisms and larger training sets, as it is well known that GPs scale very badly with the dimension of the training dataset (see e.g. Liu et al., 2018). The issue remains of the necessity of producing such large training sets, which require considerable computational resources. To face the demands in this sense for future extensions of this work, we advocate the use of Bayesian optimisation (see e.g. Frazier, 2018, for a review) to optimise the simulation of training seismograms.

We expect our methodology for waveform emulation to be improved by including physics-based information in the emulation framework, following recent work in physics-informed neural networks (PINNs; Rudy et al., 2017; E and Yu, 2017; Raissi et al., 2019; Bar and Sochen, 2019; Li et al., 2020). This is the route that has been recently followed by Smith et al. (2020) in the development of EIKONET, a deep learning solution of the eikonal equation based on PINNs. It would be interesting to apply a similar approach to our waveform emulation framework, by solving the elastic wave partial differential equation by backprojection over the network while simultaneously fitting the simulated waveforms. Such an approach might help remove



the mesh dependence of the full waveform simulations, enhancing the flexibility of our method. In addition, EIKONET has been very recently applied to hypocenter inversion (Smith et al., 2021), which makes the case even stronger for exploring further the possibility of using PINNs within the context of waveform emulation, as recently explored in 2D configurations in Moseley
et al. (2018); Moseley et al. (2020b, a) with extremely promising results.

*Code and data availability.*  Our deep learning models will be available at https://github.com/alessiospuriomancini/seismoML on publication of this paper, along with the 3D velocity model used to generate our training data.

## Appendix A: Summary of D18 emulator

Here we briefly summarise, for comparison, the surrogate model developed in D18 for fast emulation of isotropic microseismic
traces, given their source locations on a 3D grid. We report here the main steps of the procedure, referring to D18 for all details.

1.  We first compress the training seismograms, isolating in each of them the 100 dominant components in absolute values and storing their amplitudes and time indices;

2.  we then train a GP for each dominant component and for each index. Thus, in total there will be $100 \times 2 = 200$ GPs to train: each of the 100 GPs for the signal part will learn to predict the mapping between coordinates and one sorted
dominant component in the seismograms; the corresponding GP for the time index will learn to predict what is the temporal index associated to that dominant component.

3.  Once the GPs are trained, for each set of coordinates the 100 predictions for the dominant signal components and the 100 predictions for their indices will produce a compressed version of the seismogram, where the (predicted) subdominant components are set to zero.

## Appendix B: Kullback-Leibler divergence

### B1  Definition and properties

Given two probability distributions $P$ and $Q$ of a continuous random variable $X$, one possible way of measuring their distance is the Kullback-Leibler divergence (KL divergence, Kullback, 1959), which is defined as:

$$D_{\mathrm{KL}}(P||Q) = \int\limits_X p(x) \log \frac{p(x)}{q(x)} \,, \tag{B1}$$

where $p$ and $q$ are the probability densities of $P$ and $Q$, respectively. It is easy to show that $D_{\mathrm{KL}}(P||Q) \geq 0$ and that $D_{\mathrm{KL}}(P||Q) = 0 \iff P = Q$ almost everywhere: this is in line with the idea of $D_{\mathrm{KL}}(P||Q)$ being a way of measuring the distance between $P$ and





$Q$. However, we also note that that the KL divergence is not symmetric $(D_{\mathrm{KL}}(P||Q) \neq D_{\mathrm{KL}}(Q||P))$, that it does not satisfy the triangle inequality, and that it is part of a bigger class of divergences called $f$-divergences (see e.g. Gibbs and Su, 2002; Sason and Verdú, 2015; Arjovsky et al., 2017, and references therein).

**B2 Calculation of the loss function**

In Sec. 2.2.6, we introduced the KL divergence in the loss function of the Conditional Variational Autoencoder (CVAE). In that instance, we calculate $D_{\mathrm{KL}}\left(q_\theta(\boldsymbol{z}|\boldsymbol{x},\boldsymbol{c})||p(\boldsymbol{z}|\boldsymbol{c})\right)$ where both $q_\theta(\boldsymbol{z}|\boldsymbol{x},\boldsymbol{c})$ and $p(\boldsymbol{z}|\boldsymbol{c})$ are multivariate normal distributions. In particular, we choose $q_\theta(\boldsymbol{z}|\boldsymbol{x},\boldsymbol{c}) = N(\boldsymbol{z};\boldsymbol{\mu}(\boldsymbol{x},\boldsymbol{c}),\Sigma)$ and $p(\boldsymbol{z}|\boldsymbol{c}) = N(\boldsymbol{z};\boldsymbol{0},\Sigma)$, where $\Sigma$ is a diagonal matrix with all entries equal to $\sigma^2 = 0.001^2$, and $\boldsymbol{\mu}(\boldsymbol{x},\boldsymbol{c})$ is the output of the encoder network of the CVAE.

It is easy to show (Kullback, 1959; Rasmussen and Williams, 2005; Devroye et al., 2018) that the KL divergence in the case of two multivariate normal distributions reduces to

$$D_{\mathrm{KL}}\left(N(\boldsymbol{\mu}_1,\Sigma_1)||N(\boldsymbol{\mu}_2,\Sigma_2)\right) = \frac{1}{2}\log|\Sigma_2\Sigma_1^{-1}|$$
$$+\frac{1}{2}\mathrm{tr}\Sigma_2^{-1}\left((\boldsymbol{\mu}_1-\boldsymbol{\mu}_2)(\boldsymbol{\mu}_1-\boldsymbol{\mu}_2)^T + \Sigma_1 - \Sigma_2\right) . \tag{B2}$$

In our case, since $\Sigma_1 = \Sigma_2 = \Sigma$ and $\boldsymbol{\mu}_2 = 0$, we can write:

$$D_{\mathrm{KL}}\left(q_\theta(\boldsymbol{z}|\boldsymbol{x},\boldsymbol{c})||p(\boldsymbol{z}|\boldsymbol{c})\right) = \frac{1}{2}\mathrm{tr}\Sigma^{-1}\left(\boldsymbol{\mu}(\boldsymbol{x},\boldsymbol{c})\boldsymbol{\mu}^T(\boldsymbol{x},\boldsymbol{c})\right)$$
$$= \frac{1}{2\sigma^2}\sum_{i=0}^{z_{\mathrm{dim}}}\mu_i^2(\boldsymbol{x},\boldsymbol{c}) , \tag{B3}$$

where $z_{\mathrm{dim}} = 5$ is the chosen dimensionality of the latent space.

**Appendix C: Details of WGAN-GP**

In Sec. 2.2.7 we explained how standard Generative Adversarial Networks (GANs) are prone to training instabilities and 625 *mode collapse*; therefore, in this work we chose to employ a variant called Wasserstein GAN - Gradient Penalty (WGAN-GP; Arjovsky et al., 2017; Gulrajani et al., 2017). In this algorithm, two networks, called generator (G) and critic (C), are trained to minimise the Wasserstein-1 distance between the data distribution and the generative model distribution, implicitly defined by $G(\boldsymbol{z}), \boldsymbol{z} \sim p(\boldsymbol{z})$ (Arjovsky et al., 2017). The Wasserstein-1 distance is also known as the Earth Mover's distance, as it can intuitively be thought as the minimum cost to transport a certain amount of "earth" from one "pile" to another (see e.g. Rubner 630 et al., 1998, for more details). In our implementation, we additionally constrain the gradient norm of the critic's output with respect to its input to be at most one everywhere, such that the the critic lies within the space of 1-Lipschitz functions (Gulrajani et al., 2017). Finally, we include the coordinate information, and use the Kantorovich-Rubinstein duality (Villani, 2008), to express our optimisation problem as the one shown in Eq. 11.



*Author contributions.* ASM developed the theoretical framework and methodology, as well as the software implementation of the generative
models described in the paper. He also generated the synthetic data used for training, validated the models and wrote the article. DP was
involved in the implementation, validation of the models, and writing. AMGF, MPH and BJ were involved in the review and conceptualisation
of this work and contributed to writing.

*Competing interests.* No competing interests are present.

*Acknowledgements.* This work has been partially enabled by funding from Royal Dutch Shell plc and the UCL Cosmoparticle Initiative.
Some of the computations have been performed on the Wilkes High Performance GPU computer cluster at the University of Cambridge;
we are grateful to Stuart Rankin and Greg Willatt for their technical support. We wish to express our deepest gratitude to Saptarshi Das for
useful discussions and support during the initial phase of this project. We also thank Stephen Bourne, Xi Chen, Detlef Hohl, Jonathan Smith
and Teh-Ru Alex Song for useful discussions. We acknowledge the use of the software GETDIST (Lewis, 2019) for producing contour plots.
DP is supported by the STFC UCL Centre for Doctoral Training in Data Intensive Science. AMGF acknowledges funding from NERC grant
NE/N011791/1.




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
