# Peer review of "Accelerating Bayesian microseismic event location with deep learning"

_Solid Earth, 2021_

## Author Response (AR1)

**Reply to Referee 1**

We thank the referee for their positive and constructive comments. Here follow our answers to the three comments raised, in the same order as in the referee's comment:

1. We thank the referee for the comment on transfer learning, which is indeed an interesting possibility to explore for overcoming the limitation of the fixed velocity model underlying our methods. We added a paragraph to the "Discussions and conclusion" section, where we comment on the possibility of applying transfer learning to overcome the necessity of re-training our emulators from scratch for a different velocity model. We also expanded that section to comment more generally on this point, which is clearly a limitation of our method as well as other methods based on the emulation of forward models. We also note that, especially for microseismic activity in geophysical domains with velocity models that vary relatively slowly in time, re-training our models e.g. every six months on a few thousands simulated seismic traces (as the ones used in Section 3 of the manuscript) would still represent a minimal computational overhead and an enormous speed-up compared to performing Bayesian inference with simulated seismic traces at each point in the parameter space. Please see lines 595-609 of the revised manuscript in which we addressed these issues.

2. While we agree that an application on field data will be important, it goes beyond the scope of this study and it will be the subject of future work. The main reason for this is that we believe that the MS already achieves its main goal, which is to present highly-performing emulators and to showcase their superiority with respect to existing approaches. A field data application would lengthen this already long manuscript (as pointed out by the second referee) and would likely shift the reader's attention from its main point. We added a note to the "Discussions and conclusion" section (see lines 582-594 of the revised manuscript), where we explain this point and refer to planned future work where the field data application will be explored in detail. We also further clarified the main goal of the MS and that this a key step towards application of these emulation frameworks to real data. As explained in detail in the text, our approach scales much better with larger training sets than previous studies that heavily relied on Gaussian Process regression. By introducing multiple highly-performing generative models in our MS, explaining in detail their architecture and providing open source software that implements them, we are paving the way for realistic application to field data. Finally, we also discuss some details of the further work needed for real data applications (e.g., using a more realistic model of the noise characterising the seismic sources). Please see lines 582-594 of the revised manuscript in which we clarified these points.

3. We added a paragraph in the Introduction section (lines 64-72) that presents physics-based approaches to machine learning and compares them with data-driven machine learning. We also added the suggested references to the Introduction (line 69) and "Discussions and conclusion" sections (lines 612 and 614).

**Reply to Referee 2**

We thank the referee for their positive and constructive comments. Here follow our answers to the four comments raised, in the same order as in the referee's comment:

1. The model in Figure 7 is indeed heterogeneous in 3-D, but the variability on the horizontal plane is much smaller than in the vertical direction. This aspect may be hard to see by eye in Figure 7, hence we added a note in the Figure caption to explain this point. We also added some text to the manuscript explaining that although the 3-D model used has stronger variability in the vertical direction than in the horizontal plane our method is general and applicable for stronger 3D heterogeneity (see lines 595-609 of the revised manuscript, also in connection with the next point).

2. The deep learning emulators need to be trained on synthetics generated for a given fixed velocity model. Hence, for a different velocity model, the same architecture will perform differently. A new optimisation of the emulator training should be performed for every new model considered, hence it is not a priori easy to predict the performance of the models trained in this work on a more heterogeneous velocity model. For moderately more heterogeneous models, the networks developed in this paper should perform well, albeit not optimally. On the other hand, for models with much stronger heterogeneity, an increased number of training samples will likely be needed to obtain an accurate emulator. Nevertheless, as explained in our reply to referee 1, given the speed-up of our approach, such new optimisations should be computationally feasible. We modified the manuscript to clarify this (see lines 595-609 in the "Discussion and conclusions" section of the revised manuscript).

   As for the applicability of our approach to real data, as explained in our reply to referee 1, we note that the framework developed in our paper represents an important starting point towards real data applications. Nevertheless, in order to enhance the performance in real data applications some additional work is needed, notably a more realistic model for the noise characterising the seismic sources. We highlight that even though this represents an additional computational requirement, once such a model has been obtained it is straightforward to insert it in the pipeline developed in the paper. The modular structure of our software allows for seamless integration of any type of noise for the seismic sources. We added a few paragraphs in the "Discussion and conclusions" section to explain these various points (see lines 582-594 of the revised manuscript).

3. We added explanations to the manuscript about some details of the implementation, avoiding statements unsupported by quantitative estimates. Please see lines 135-137, 180-181, 188-189, 229 of the revised manuscript.

4. We performed several tests with and without using the distance d and found that including it helps the Gaussian processes trained to learn the amplitude and time shift coefficients in each generative model, since those two coefficients depend strongly on the distance of each seismic trace from the receiver. We added a comment on this in sections 3.1 (lines 141-143) and 3.2 (lines 355-357).